# Design of Fragrance Formulations with Antiviral Activity Using Bayesian Optimization

**DOI:** 10.3390/microorganisms12081568

**Published:** 2024-07-31

**Authors:** Fan Zhang, Yui Hirama, Shintaro Onishi, Takuya Mori, Naoaki Ono, Shigehiko Kanaya

**Affiliations:** 1Material Science Research, Kao Corporation, 1334 Minato, Wakayama-shi 640-8580, Wakayama, Japan; zhang.fan2@kao.com; 2Division of Information Science, Graduate School of Science and Technology, Nara Institute of Science and Technology, 8916-5 Takayama-cho, Ikoma 630-0192, Nara, Japan; nono@is.naist.jp; 3Biological Science Research, Kao Corporation, 2606 Akabane, Ichikai-machi, Haga-gun 321-3426, Tochigi, Japan; hirama.yui2@kao.com (Y.H.); oonishi.shintarou@kao.com (S.O.); mori.takuya@kao.com (T.M.); 4Data Science Center, Nara Institute of Science and Technology, 8916-5 Takayama-cho, Ikoma 630-0192, Nara, Japan

**Keywords:** antiviral activity, fragrance, Bayesian optimization

## Abstract

In case of future viral threats, including the proposed Disease X that has been discussed since the emergence of the COVID-19 pandemic in March 2020, our research has focused on the development of antiviral strategies using fragrance compounds with known antiviral activity. Despite the recognized antiviral properties of mixtures of certain fragrance compounds, there has been a lack of a systematic approach to optimize these mixtures. Confronted with the significant combinatorial challenge and the complexity of the compound formulation space, we employed Bayesian optimization, guided by Gaussian Process Regression (GPR), to systematically explore and identify formulations with demonstrable antiviral efficacy. This approach required the transformation of the characteristics of formulations into quantifiable feature values using molecular descriptors, subsequently modeling these data to predict and propose formulations with likely antiviral efficacy enhancements. The predicted formulations underwent experimental testing, resulting in the identification of combinations capable of inactivating 99.99% of viruses, including a notably efficacious formulation of five distinct fragrance types. This model demonstrates high predictive accuracy (coefficient determination $R_{\mathrm{cv}}^{2}$ > 0.7) and suggests a new frontier in antiviral strategy development. Our findings indicate the powerful potential of computational modeling to surpass human analytical capabilities in the pursuit of complex, fragrance-based antiviral formulations.

## 1. Introduction

Severe acute respiratory syndrome coronavirus (SARS-CoV)-2 infection caused a global pandemic that began in March 2020 and had a significant impact on human lives. Over the past 150 years, a series of pandemics and endemics have emerged, including the Russian influenza, Spanish flu, Asian influenza, Hong Kong influenza, 2009 H1N1 influenza, SARS-CoV epidemic, and Middle East respiratory syndrome [1,2]. Hence, the anticipation of a forthcoming pandemic, potentially triggered by an unknown virus such as Disease X, persists. Nevertheless, there are still a few effective countermeasures against pandemics, emphasizing the urgent need for their development.

Countermeasures for the Disease X pandemic encompass both curative and preventive strategies. One avenue for curing diseases involves drug administration; nevertheless, developing such drugs necessitates a significant financial investment and considerable time [3]. Vaccination is an effective preventive method, yet it carries the potential for causing side effects [4].

Therefore, we attempted to develop a safe and simple method for preventing viral infections by focusing on fragrance molecules as antiviral substances. Fragrances are generally considered safe since they find widespread use in various aspects of our daily lives, and inhaling their vapors has been utilized as a remedy for specific illnesses [5]. The identification and use of fragrances with high antiviral activity could potentially lead to the inactivation of viruses on the body, clothes, living spaces, and public spaces. This could be achieved by incorporating such fragrances into detergents or other products. Additionally, spraying these fragrances into the air may help combat airborne viruses. Some fragrances, such as thymol, possess antiviral activity [6]; however, the relationship between their chemical structure and antiviral activity remains unclear. Recently, Swarit et al. developed a machine-learning model using approximately 200 data points on the antiviral activity of fragrances [7]. They calculated the importance of substructures for antiviral activity using Shapley Additive exPlanations (SHAP) [8] in their model. They also successfully identified new fragrances with high antiviral activity. Nevertheless, the fragrances discovered so far have been inappropriate for social implementation in terms of concentration and fragrance as well as virus inactivation effect. The fragrance concentration must be sufficiently high to achieve the desired effect. Nonetheless, increasing fragrance concentrations can raise safety concerns. Wani et al. reported that essential oils containing diverse compounds have high anti-influenza and anti-coronaviral activities [9]. The IC_50_ for H1N1 influenza virus is reported to be <3.1 μL/mL for several essential oils and 2.6 μg/mL for carvacrol, a constituent of essential oil. This suggests that by optimally combining the active components within essential oils, it may be possible to identify highly active combinations at lower concentrations. Additionally, some fragrances, such as thymol, can enhance antimicrobial activity when combined with other fragrances, albeit against bacteria [10]. Consequently, we consider that the antiviral activity can be enhanced by appropriately combining fragrances. Yet, finding the optimal combination of fragrances can be a challenging task due to the vast number of possible combinations and the complex interactions between components. Bayesian optimization based on the Gaussian Process Regression (GPR) is used to efficiently explore vast spaces. This efficient search method is used in many fields, such as material development and drug discovery [11]. In Quantitative Structure–Activity Relationship (QSAR), compounds are often vectorized using 208 RDKit descriptors [12], ECFP4 [13], or similar methods to build machine learning models. Nonetheless, there are few examples of Bayesian optimization for formulations that combine compounds without fixing the number of compounds to be blended, and vectorization of the formulations is not clear. Moreover, the vast number of combinations makes it nearly impossible to optimize this formulation solely relying on human experience and intuition. Hence, we collected and analyzed a total of 173 fragrances in search of potential enhancements in antiviral activity. This effort aimed to develop fragrance formulations with superior antiviral properties. We aimed to develop a method for preventing viral infections by using fragrance molecules as antiviral substances.

## 2. Materials and Methods

### 2.1. Data

We gathered a total of 173 diverse fragrance compounds comprising alcohols, aldehydes, esters, ethers, and other chemical compounds (Appendix A). Many of the fragrances were purchased from reagent manufacturers such as FUJIFILM Wako Pure Chemical Corporation; Tokyo Chemical Industry Co., Ltd. (Tokyo, Japan); and Sigma-Aldrich (St. Louis, MO, USA). For those not available from these reagent manufacturers, we sourced them from companies like Bedoukian Research, Inc. (Danbury, CT, USA); International Flavors & Fragrances Inc. (New York, NY, USA); and Givaudan S.A. (Vernier, Switzerland). Not all the utilized fragrance compounds were of high purity. If the fragrance compound was a mixture, it was treated as if only the main component was present. Subsequently, we determined 1651 viral infectious titers for their formulations, which included one fragrance (173 formulations), two fragrances (879 formulations), and three fragrances (599 formulations). Regardless of the purity of the fragrance compounds used, in the research, fragrance compound concentrations were uniformly evaluated at 0.1 vol%. For formulations containing two or three types of compounds, the concentrations of each compound were equally allocated. For instance, in a formulation composed of compounds A and B, each was prepared at 0.05 vol%; similarly, for a formulation consisting of compounds A, B, and C, each was prepared at 0.033 vol%. Formulations with the composition A, A, and B were also designated as formulations with three types of fragrances within the scope of this research. Various formulations using two fragrances were selected and evaluated; formulations with three fragrances by adding one additional fragrance to the formulations with two fragrances with high antiviral activity were evaluated. Multiple evaluations were conducted for each formulation, ranging from one to five times, and the average of the measured values was utilized as the supervised data.

### 2.2. Measurement of Antiviral Activity

Influenza virus type A (A/Puerto Rico/8/1934, H1N1) was used as a test virus. A solution of 6 vol% compounds and 60 vol% dipropylene glycol (FUJIFILM Wako Pure Chemicals Co. Ltd., Osaka, Japan) combined with 34 vol% of UltraPure DNase/RNase-free distilled water (Thermo Fisher Scientific Japan K.K., Kanagawa, Japan) was used as the stock solution. The stock solution was stored at 4 °C until use in Proteosave tubes (Sumitomo Bakelite Co., Ltd., Tokyo, Japan) and diluted 30 times with serum-free medium (Hybridoma-SFM; Thermo Fisher Scientific). Subsequently, 60 μL of the solution was added to a 96-well Proteosave plate (Sumitomo Bakelite), and 60 μL of virus solution (7.8 × 10^5^ FFU) was added and allowed to react for 30 min at room temperature (approximately 23 °C) at a final concentration of 0.1 vol% of the compound. When combining multiple compounds, the total concentration was standardized to 0.1 vol%, for example, 0.05 vol% for each of the two species, and subsequently reacted with the virus. After the reaction, the virus was diluted 10- to 10^4^-fold with SFM and used to infect MDCK cells (derived from canine renal tubular epithelial cells) that were previously cultured in 12- or 48-well plates. The viral titer was determined by a Focus Forming Assay [14]. Specifically, the viral infection titer was measured using the number of foci formed after approximately 18 h of incubation at 37 °C and 5 vol% CO_2_. The infectious titer, when reacted with a control 1 vol% dipropylene glycol solution, was set at 100%, and the percentage of the viral infectious titer for each formulation was calculated. A lower viral infection titer indicated higher antiviral activity. The determination of the cytotoxicity of the compounds used was based on the assessment of cell detachment in MDCK cells in the Focus Forming Assay. For instance, a formulation that exhibited approximately 2-log antiviral activity was subjected to the Focus Forming Assay using a solution diluted more than 1000-fold after the reaction with the virus. In this case, even the solution with a higher residual compound concentration, diluted by 100-fold, was subjected to the Focus Forming Assay, and it was confirmed that there was no detachment of MDCK cells, and the virus was detected as foci, leading to the conclusion of no cytotoxicity. At the test concentrations used in this study, no compounds were found to exhibit antiviral activity due to cytotoxicity to the MDCK cells.

### 2.3. Descriptors for Fragrance Formulations

The fragrances were represented using 208 RDKit descriptors. Fragrances of relatively low purity were vectorized using the chemical structures of their main components. The formulations were expressed using statistics on the RDKit descriptors of each formulated fragrance, such as the weighted mean, weighted standard deviation, maximum, and minimum. Statistics were calculated using the following equation:(1)$$x_{i,\;\mathrm{mean}}=\sum_{n}C_{n}x_{n,i}$$
(2)$$x_{i,\;\mathrm{std}}=\sqrt{\sum_{n}C_{n}{\left(x_{n,i}-x_{i,\;\mathrm{mean}}\right)}^{2}}$$
(3)$$x_{i,\max}=\max\{x_{n,i}|n\text{∈}\mathrm{formulation}\}$$
(4)$$x_{i,\min}=\min\{x_{n,i}|n\text{∈}\mathrm{formulation}\}$$
where $x_{n,i}$ indicates the value of the i-th RDKit descriptor of the n-th fragrance in the formulation, and $C_{n}$ indicates the concentration of the n-th fragrance in the formulation.

### 2.4. Gaussian Process Regression (GPR)

GPR is a nonparametric stochastic model that can help calculate not only the predicted mean but also the predicted standard deviation [15]. The prior distribution is defined through a mean function and a covariance function via kernel functions. Typically, zero serves as the mean function. The covariance function can be computed as follows, utilizing the mean and covariance functions:(5)$$p\left(y^{*}|x^{*},y,\;X\right)=N\left(k_{*}^{T}K^{-1}y,\;k_{**}-k_{*}^{T}K^{-1}k_{*}\right)$$

Here, **X** and **y** represent the explanatory variables and the target variable in the training data points, respectively, while $x^{*}$ and $y^{*}$ denote the explanatory variables and the predicted value for the data to be forecasted. $x_{i}$ represents the i-th data point of the training data **X**. k represents the kernel function, $k_{*}$ represents the vector where the i-th element is $k(x_{i},x^{*})$, $k_{**}$ represents $k(x^{*},x^{*})$, and K represents the matrix where the i,j-th element is $k(x_{i},x_{j})$. Nonlinear kernel functions enable the GPR to make nonlinear predictions. Therefore, the following two types of kernel functions were used in this study:(6)$$k_{\mathrm{RBF}}\left(x^{(i)},x^{(j)}\right)=\theta_{0}\cdot \exp\left(-\frac{{d_{i,j}}^{2}}{2\theta_{1}^{2}}\right)+\theta_{2}\cdot \delta \left(i,j\right)$$
(7)$$k_{\nu }\left(x^{(i)},x^{(j)}\right)=\theta_{0}\cdot \frac{2^{1-\nu }}{\Gamma \left(\nu \right)}{\left(\frac{\sqrt{2\nu }d_{i,j}}{\theta_{1}}\right)}^{\nu }K_{\nu }\left(\frac{\sqrt{2\nu }d_{i,j}}{\theta_{1}}\right)+\theta_{2}\cdot \delta \left(i,j\right)$$

Here, $d_{i,j}$ indicates the Euclidean distance between $x^{(i)}$ and $x^{(j)}$, $\theta_{n}$ indicates hyperparameter that takes on non-negative values, $K_{\nu }$ indicates a modified Bessel function, Γ indicates gamma function, and $\delta \left(i,j\right)$ indicates Kronecker’s δ. In $k_{\nu }$, the parameter ν controls the smoothness of the model, and as ν approaches infinity, $k_{\nu }$ converges to $k_{\mathrm{RBF}}$. When using $k_{\nu }$ kernel function, we chose ν=1.5 in this study. We tuned $\theta_{n}$ of kernel functions to maximize the log marginal likelihood, and it is defined as follows:(8)$$\log p\left(y|X\right)=-\frac{1}{2}y^{T}K^{-1}y-\frac{1}{2}\log\left|K\right|+C$$

C indicates a constant that does not depend on **X** and **y**.

### 2.5. Feature Selection

Feature selection was conducted before modeling through the following process: First, we removed variables from the dataset in which all data points had the same value; then, we computed the correlation between the remaining variables and deleted one feature from each pair, displaying a correlation coefficient of 1.0.

### 2.6. Building and Evaluating Models

A 10-fold cross-validation approach was used to assess the predictive accuracy of the models, considering different kernel functions, preprocessing types, and descriptor types. In cases where the training data contained variables with identical values, these redundant variables were excluded. We used the coefficient of determination ($R_{\mathrm{cv}}^{2}$), mean absolute error (${\mathrm{MAE}}_{\mathrm{cv}}$), root mean squared error (${\mathrm{RMSE}}_{\mathrm{cv}}$), and mean absolute percentage error (${\mathrm{MAPE}}_{\mathrm{cv}}$) as the evaluation metrics. $R_{\mathrm{cv}}^{2}$ indicates how well the prediction model fits; ${\mathrm{RMSE}}_{\mathrm{cv}}$ and ${\mathrm{MAE}}_{\mathrm{cv}}$ indicate the magnitude of the prediction error; ${\mathrm{MAPE}}_{\mathrm{cv}}$ indicates the ratio of the original value to the prediction error. These metrics were calculated using the following equations:(9)$$R_{\mathrm{cv}}^{2}=1-\frac{\sum_{i}{\left(y^{\left(i\right)}-y_{\mathrm{CV}}^{\left(i\right)}\right)}^{2}}{\sum_{i}{\left(y^{\left(i\right)}-y_{\mathrm{mean}}\right)}^{2}}$$
(10)$${\mathrm{RMSE}}_{\mathrm{cv}}=\sqrt{\frac{\sum_{i}{\left(y^{\left(i\right)}-y_{\mathrm{CV}}^{\left(i\right)}\right)}^{2}}{n}}$$
(11)$${\mathrm{MAE}}_{\mathrm{cv}}=\frac{\sum_{i}\left|y^{\left(i\right)}-y_{\mathrm{CV}}^{\left(i\right)}\right|}{n}$$
(12)$${\mathrm{MAPE}}_{\mathrm{cv}}=\frac{\sum_{i}\left|(y^{\left(i\right)}-y_{\mathrm{CV}}^{\left(i\right)})/y^{\left(i\right)}\right|}{n}\text{×}100$$

Here, $y^{\left(i\right)}$ is the y value of the i-th sample, $y_{\mathrm{CV}}^{\left(i\right)}$ is the predicted value of the i-th sample in the 10-fold cross-validation, and $y_{\mathrm{mean}}$ is the average of y. When building the GPR model, if y was transformed by a log or logit function, the predicted value was transformed by its inverse function and used as $y_{\mathrm{CV}}^{\left(i\right)}$.

### 2.7. Genetic Algorithm (GA)

GA is a metaheuristic optimization algorithm [16]. The formulation design is a combination problem. Thus, we applied GA to optimize the formulation by maximizing the acquisition function. GA was executed several times with different random states using the tournament selection method, featuring a tournament size of three, a population size of 300, 100 generations, a crossover rate of 0.5, and a mutation rate of 0.2. We experimentally evaluated several formulations with large acquisition functions, as determined by GA.

### 2.8. Software and Implementation

All regression models were calculated using Scikit-learn (version: 0.23.2) [17]. The RDKit descriptors were computed using RDKit (version: 2021.09.1) [12]. GA was conducted using DEAP (version: 1.3.1) [18]. All other data analyses were performed using Python 3.7. 

## 3. Results

In this study, we first analyzed the acquired training data for the relationship between structure and antiviral activity. Then, each fragrance molecule was represented using RDKit descriptors, and the formulation was expressed using statistics such as the mean of each descriptor of the formulated fragrance to build machine learning models. This approach enables the representation of all fragrance formulations with vectors of the same length, allowing them to be treated within a single model. We first evaluated the antiviral activity values of 173 single molecules, 879 formulations using combinations of two, and 599 formulations using combinations of three. Using these data, we constructed the GPR and proposed formulations with large acquisition functions using the constructed model and verified the prediction accuracy by experimental evaluation. Subsequently, we identified formulations that exceeded the highest level of antiviral activity.

### 3.1. Relationship between Fragrances and Viral Infectivity Titers

First, the relationship between the chemical structure of a single fragrance compound and its viral infectivity titers was investigated. Viral infectivity titers were aggregated based on the presence of certain substructures and represented using box plots, as shown in Figure 1. Six substructures were employed: alkyl alcohol, alkyl acid, aldehyde, ester, ether, and phenol. The presence or absence of these substructures in fragrance molecules was determined by whether six descriptors—fr_Al_OH, fr_Al_COO, fr_aldehyde, fr_ester, fr_ether, and fr_phenol—defined in RDKit were greater than zero or equal to zero. In our evaluation system, compounds with aldehyde and phenolic structures tended to exhibit antiviral effects more readily in isolation, whereas compounds containing alkyl alcohol, carboxylic acid, ether, or ester structures were less likely to demonstrate high virus inactivation effects on their own. 

Next, we considered the virus inactivation effect of formulations with two types of fragrance molecules. Initially, we investigated how the antiviral efficacy of formulations changed based on whether they included aldehyde compounds that were individually found to have high antiviral effects (Figure 2). The results showed that combinations containing aldehydes exhibited higher median values. 

Furthermore, we compared the virus inactivation effects of single aldehyde compounds with those of formulations containing aldehydes and another type of fragrance compound (Figure 3). 

The formulations containing two types of fragrances achieved higher maximum viral infectivity titers than those observed with aldehyde compounds in isolation. Since the total concentration of fragrance compounds was unified at 0.1 vol% in this study, we confirmed that certain combinations exhibit synergistic effects. Additionally, we created box plots based on the types of compounds combined with aldehyde compounds in two-component formulations (Figure 4). As a result, the median of −log (viral infectivity titers) + 2, which equivalent to the log reduction rate, for the aldehyde compounds alone was approximately 0.35, while the medians of –log (viral infectivity titers) + 2 for the combination of aldehyde compounds with aldehydes or phenolic compounds were approximately 0.70. This indicates that the combination with aldehydes or phenolic compounds yields promising results. 

Specific examples of each are provided: a combination of syringa aldehyde, which has a formyl group at the benzyl position, and trans-2-undecenal for a combination of aldehydes and aldehydes combination; and thymol combined with cis-4-decenal for a combination of aldehydes and phenols, both of which exhibited higher activity than when used individually (Figure 5). Interestingly, adding trimethyl hexanol, which does not have a high antiviral effect on its own, to the thymol and cis-4-decenal mixture resulted in an even greater antiviral effect (Figure 6).

### 3.2. Evaluating GPR Performances with Type of Kernel Functions, Preprocessing, and Descriptor

GPRs with various kernel functions described in the Methods Section were built to predict the viral infectious titers of fragrance formulations using statistics on the RDKit descriptors of each formulated fragrance. The prediction accuracies were also compared using the case in which a log or logit transformation was performed on y as a preprocessing step. The log and logit transformations are defined as follows:(13)$$y^{\prime }=\log_{10}y$$
(14)$$y^{\prime }=\ln\frac{y}{100-y}$$

The shape of the functions is illustrated in Figure 7. Both the logarithmic and logit functions exhibit steep slopes around the value of 0, which implies that when the value of y is near 0, even small predictive errors are heavily penalized. Consequently, by transforming y before training a machine learning model to minimize error, the model is effectively trained to reduce the predictive error around the value of 0. Additionally, because the logit function also has a steep slope around the value of 100, training a machine learning model after performing a logit transformation will similarly result in the model being trained to minimize predictive errors around the value of 100.

During the application of the logit transformation, viral infectious titers exceeding 100 were considered outliers and were excluded from the training data. Since the number of samples with viral infection titers >100 was very small, their omission had a negligible effect on the models. Histograms of viral infection titers are shown in Figure 8. In the plot shown in Figure 8a, the data were observed to be clustered near the value of 0. However, upon applying logarithmic and logit transformations, the data points became more dispersed, as demonstrated in Figure 8b,c, thereby making the differences in y values near 0 more pronounced. Within the dataset, there were 675 data points with y ≤ 10 (equivalent to a 1-log reduction), 287 data points with y ≤ 1.0 (equivalent to a 2-log reduction), and 8 data points with y ≤ 0.1 (equivalent to a 3-log reduction).

The prediction performance of the GPRs using only the weighted mean as the statistical descriptor is shown in Table 1. A high $R_{\mathrm{cv}}^{2}$ value indicates good predictive accuracy of the model, whereas lower values of ${\mathrm{MAE}}_{\mathrm{cv}}$, ${\mathrm{RMSE}}_{\mathrm{cv}}$, and ${\mathrm{MAPE}}_{\mathrm{cv}}$ signify better predictive accuracy of the model. Regardless of the kernel function used for GPR, $R_{\mathrm{cv}}^{2}$ was the highest when no transformation was performed on y as a preprocessing step; ${\mathrm{MAPE}}_{\mathrm{cv}}$ was also high. In contrast, when log transformation was applied to y for preprocessing, ${\mathrm{MAPE}}_{\mathrm{cv}}$ was the lowest; $R_{\mathrm{cv}}^{2}$ was also low. When logit transformation was applied to y as a preprocessing step, $R_{\mathrm{cv}}^{2}$ was high, and ${\mathrm{MAPE}}_{\mathrm{cv}}$ was low, indicating that the model was well balanced. Regardless of the preprocessing, the most accurate predictions were made when $k_{1.5}$ was used as a kernel function. Table 2 shows the evaluation results of the GPRs using not only the weighted mean but also the weighted standard deviation as statistical descriptors. The prediction performance was better when using the weighted mean and weighted standard deviation as statistical descriptors than when using the weighted mean alone, regardless of which kernel function was used or how it was preprocessed. Table 3 shows the evaluation results of the GPRs using the weighted mean, weighted standard deviation, and maximum and minimum as the statistical descriptors. The predictive performance was higher when the maximum and minimum values were also utilized as statistical descriptors. The Y-Y plots for all predictions by GPR with $k_{1.5}$ using the weighted mean, weighted standard deviation, maximum, and minimum as statistical descriptors are shown in Figure 9.

### 3.3. Validation of Bayesian Optimization Performance Using Statistical Descriptors

We validated whether statistical descriptors with feature selection and GPR could efficiently identify formulations with better values. The simulation was performed using a dataset of formulations that included measured viral infection titers. In this experiment, GPR with $k_{1.5}$ using the weighted mean, weighted standard deviation, and maximum and minimum as statistical descriptors had the highest prediction performance in terms of $R_{\mathrm{cv}}^{2}$ and ${\mathrm{MAPE}}_{\mathrm{cv}}$ for this dataset. Viral infection titer values were logit-transformed and multiplied by −1 and used. Seventy data points from formulations containing one type of fragrance and 30 data points from formulations containing two types of fragrances were randomly selected as initial training data. GPR with a $k_{1.5}$ kernel was trained with training data, and the calculated expected improvement (EI) was defined, as below, as the acquisition function using the constructed GPR for data not in the training data.
(15)$$\mathrm{EI}\left(x\right)=\int_{y_{\max}+\varepsilon }^{\infty }t\cdot \frac{1}{\sqrt{2\pi \sigma^{2}\left(x\right)}}\exp\left\{-\frac{{\left(t-\mu \left(x\right)\right)}^{2}}{2\sigma^{2}\left(x\right)}\right\}\mathrm{dt}\;$$
where $\mu \left(x\right)$ and $\sigma^{2}\left(x\right)$ indicate predicted mean and standard deviation, and 0.01 multiplied by the standard deviation of target variable in training data was used as ε. Subsequently, we added the formulation with the largest EI from approximately 1600 formulations not yet included in the training data. This process was repeated 100 times to verify the change in the maximum target value. The outcomes were contingent on the initial training data; hence, we changed the initial dataset and conducted 100 experiments. The results showed that the descriptor and GPR could be used to efficiently propose formulations with high virus inactivation efficacy (Figure 10). Furthermore, out of 100 searches, formulations containing one type of fragrance were selected 1.8 times, those with two types of fragrances were selected 12.6 times, and those with three types of fragrances were selected 85.6 times on average. This indicates that, although formulations containing only up to two types of fragrances were used in the training data, more complex formulations could be appropriately explored. 

### 3.4. Virtual Screening and Experimental Validation

In this study, with the objective of identifying formulations with a higher antiviral activity, we constructed a GPR with a $k_{1.5}$ kernel using all data. We used the weighted mean, weighted standard deviation, and maximum and minimum as statistical descriptors. In terms of ${\mathrm{MAE}}_{\mathrm{cv}}$, log transformation as preprocessing was more favorable, whereas in terms of ${\mathrm{MAPE}}_{\mathrm{cv}}$ and $R_{\mathrm{cv}}^{2}$, the highest values were observed when applying logit transformation. Accordingly, we trained two GPRs: One was trained on the data for which y was log-transformed and multiplied by −1 as a preprocessing step, and the other was trained on the data for which y was logit-transformed and then multiplied by −1. Subsequently, we calculated the EI for each model and optimized the formulation to maximize the product of two EIs. Up to this point, we evaluated formulations comprising a maximum of three types of fragrances. Next, we specifically sought formulations that included three, four, or five types of fragrances to maximize the product of two EIs. However, the number of possible formulations containing four or five types of fragrances selected from 173 exceeded 10^7^, and calculating the product of two EIs for all formulations was not possible. Therefore, we used the GA to search for three formulations containing four types of fragrances and three formulations containing five types of fragrances with a higher product of two EIs, and their virus infection titer was evaluated by experiments. The results are shown in Figure 11. The viral infectivity titers for formulations containing five types of fragrances were 0.0066 (about 4.2-log reduction), 0.15 (about 2.8-log reduction), and 2.6 (about 1.6-log reduction). Formulations containing four types of fragrances exhibited viral infectivity titers of 0.046 (about 3.3-log reduction), 0.058 (about 3.2-log reduction), and 0.42 (about 2.4-log reduction). Formulations containing three types of fragrances exhibited viral infectivity titers of 33.7 (about 0.47-log reduction), 51.2 (about 0.29-log reduction), and 39.7 (about 0.40-log reduction). The formulations containing four and five types of fragrances exhibiting high antiviral activity are presented in Figure 12 and Figure 13.

## 4. Discussion

### 4.1. Comparison of the Results of This Evaluation System with Those of Other Studies

First, a comparison was conducted between this evaluation system and other studies regarding the relationship between the chemical structures of single-fragrance compounds and viral infectivity titers. Regarding alkyl alcohols, it is well known that alcohol compounds such as ethanol are commonly used as disinfectants. However, under the evaluation conditions of this study, they did not exhibit significant virus inactivation effects. This is thought to be due to the insufficient concentration of the ethanol used in the evaluation, which was at 0.1 vol%. Subsequently, considering that citric acid has been reported to have inactivation capabilities against the influenza virus [19], it was anticipated that alkyl acids would demonstrate a high virus inactivation effect. Nonetheless, the results indicated that the presence of alkyl acid did not significantly affect the virus inactivation efficacy. Previous studies reported findings at a pH of 3.2, and it is conceivable that the virus inactivation effect did not manifest in our system because the concentrations of fragrances were low, and the pH was not as acidic. Furthermore, formaldehyde and glutaraldehyde were reported to possess virus inactivation effects [20], prompting a comparison of the impact of aldehyde substructures on virus inactivation. The results revealed that aldehyde compounds had a higher virus inactivation effect than those without, which is consistent with prior studies. Other typical substructures, such as esters and ethers, were also compared, but no significant effect on virus inactivation was observed based solely on the presence of specific substructures. Given the scarcity of reports on ethers and esters contributing to virus inactivation, these results are likely not unique to our evaluation system. However, fragrance molecules containing phenolic structures did show a difference in median virus infectivity titer compared to those without phenolic structures. Green tea extracts, which are rich in polyphenols such as catechins, have been reported to inactivate influenza viruses [21,22], and epigallocatechin-3-gallate, a type of polyphenol, is considered to interact with viral hemagglutinin on the surface, inhibiting adsorption to host cells and leading to inactivation [23]. These findings agree with the results of our evaluation system and suggest that phenolic hydroxyl groups contribute to virus inactivation. In our evaluation system, the functional groups that demonstrated antiviral activity were those previously reported for their antiviral activity, whereas no substituents exhibited antiviral effects exclusively in our system. Consequently, it is reasonable to consider that our evaluation system represents a stringent testing environment.

Regarding combinations of fragrances, there was a tendency for combinations of compounds with different substructures or skeletons to produce synergistic effects. Syringa aldehyde and trans-2-undecenal, as shown in Figure 5a, are both aldehydes, and the difference between the aliphatic and benzyl aldehydes in their electron states and molecular shapes may have resulted in different interactions with the lipid bilayer, thereby leading to synergistic effects. The underlying mechanism of the synergistic effect observed in the combination of thymol and aldehydes, as shown in Figure 5b, can be considered as follows. Thymol is considered a defect-forming agent [24], but defect formation alone may not sufficiently inactivate the virus. It is hypothesized that the aldehyde penetrates the virus through the pores formed by thymol and rapidly damages the RNA, thus resulting in the synergistic effect. Future research will investigate the impact of each fragrance compound on the viral lipid bilayer and will conduct detailed analyses.

### 4.2. Model Prediction Performance

In this study, we constructed GPR models with various kernel functions to predict the viral infectivity titers of formulations. The models were built using the viral infectivity titers as the target variable and were evaluated through cross-validation, where we observed that the MAPEs exceeded 200. However, when a logarithmic or logit transform was applied, the errors near zero were smaller, and the MAPE was below 100, greatly improving the prediction error rate, as intended. In addition, three sets of formulations, each consisting of three, four, and five components and exhibiting high EI product, were assessed using the GPR models developed through the application of log and logit transformations. Consequently, multiple formulations with high antiviral activity were identified, corroborating the model’s high predictive accuracy. However, the evaluation of the antiviral efficacy of formulations containing the three fragrances proposed by Bayesian optimization demonstrated low effectiveness. This outcome is likely due to the considerable difficulty in designing formulations that surpass the previously best-achieved values when combining the three fragrances, leading to the exploration of extrapolations that are challenging to predict accurately. Furthermore, during this evaluation, although the formulation with the highest antiviral efficacy was a formulation with five types of fragrances, the mean of the viral infectivity titer was lower in the formulation with four types of fragrances. This observation suggests that the models may have difficulty predicting the viral infectivity titer of a formulation as the number of components increases. However, this issue could potentially be addressed by experimentally evaluating the viral infectivity titer of formulations with many types of fragrances, incorporating these data into the training dataset, and retraining the models.

The formulations with high antiviral activity identified using machine learning also included combinations of aldehydes and thymol, similar to previous formulations with two or three types of fragrances. This suggests that the GPR models learned that thymol and aldehydes are essential components for high antiviral activity. Interestingly, the other fragrances incorporated into these formulations were those that individually exhibited low antiviral activity, with viral infectivity titers of 50 (about a 0.3-log reduction) or higher. These formulations included a variety of fragrances, such as branched alcohols, alicyclic compounds with esters, cyclic ketones, and cyclic ethers. Designing such formulations manually would be nearly impossible; they were discovered through the application of machine learning techniques.

### 4.3. Usefulness of Virus Inactivation by Combinations of Fragrances

Our data showed a tendency for viral infection titer reduction by appropriately increasing the number of types of fragrances in the formulation. This phenomenon could be attributed to the distinct virus inactivation mechanisms exhibited by each flavoring agent, and their combination potentially led to a synergistic effect. Essential oils containing multiple fragrances are known to exert virus-inactivating effects [9]. This study demonstrated that the IC_50_ for influenza virus was <3.1 μL/mL for several essential oils. Due to the differing specific gravities of the compounds, a simple concentration comparison is not feasible. However, we formulated 2–5 compounds at a 0.1% concentration and tested them against the influenza virus, observing more than a 2-log reduction in viral activity in several formulations. Thus, our formulations demonstrate activity that is comparable to or even superior to existing findings. This is also thought to be because multiple fragrances act on viruses through different mechanisms. However, essential oils are not specifically designed to inactivate viruses and cannot be used at high concentrations because of safety concerns. The formulation found in this study not only inactivates viruses much more effectively than essential oils but may also be safer because it uses only compounds used as fragrances. Verifying virus inactivation using compounds in space remains challenging [14], and in using this method to optimize not only the type of fragrances in the formulation but also the concentration and scent of formulation, further research is necessary to apply the combinations identified in this study for the control of airborne viruses. Firstly, it is essential to achieve the concentrations that demonstrate antiviral activity in environments where viruses are present. Specifically, in the air, it is necessary to reach a 0.1% concentration in aerosols or droplets. This requires that the spatial concentration of the fragrance achieves the effective concentration. To achieve this effective concentration, methods such as diffusers or sprays must be utilized to volatilize the fragrance, and spatial concentrations need to be measured. Additionally, it is necessary to measure the persistence of these concentrations in inactivating airborne viruses and determine that they do not affect safety or cause allergy risks for applicable use. By advancing these investigations, it will be possible to disperse fragrances in the air to control viruses in real-world environments. 

While formulations with antiviral activity can be identified using machine learning, this approach does not elucidate the underlying mechanisms. Investigating these mechanisms is crucial not only for advancing scientific research but also for accurately assessing their impact on human health and the environment. Moreover, a deeper understanding of these mechanisms facilitates the application of these formulations in various contexts, such as disinfectants, hygiene products, and medical settings, making it equally important from an application perspective. Even when studying how a single compound inactivates a virus, the focus is solely on the interaction between the virus and the compound, which is still challenging. However, when examining how a formulation with two compounds inactivates a virus, the complexity significantly increases. It becomes necessary to consider multiple interactions: between compound A and the virus, compound B and the virus, and between compound A and compound B. This added complexity greatly elevates the research difficulty. Moving forward, we plan to tackle this challenging issue, and we believe that efficiently developing highly safe virus inactivation agents is possible in anticipation of future pandemics.

## 5. Conclusions

In this study, we identified fragrance formulations with high antiviral activity using GPR models and virtual screenings with EIs product. We first created a training dataset by measuring viral infectious titers of formulations containing one to three types of fragrances. Using statistics on the RDKit descriptors for explanatory variables, we developed highly accurate GPR models. With these models, we performed virtual screenings to find formulations with four or five types of fragrances. Experimental evaluation revealed a formulation with five fragrances that showed superior antiviral activity at low concentrations. The formulation is anticipated to have the potential for practical application in the prevention and control of infectious diseases, with future practical implementation expected. This methodology offers a valuable approach for addressing emerging pandemics.

## Figures and Tables

**Figure 1 microorganisms-12-01568-f001:**
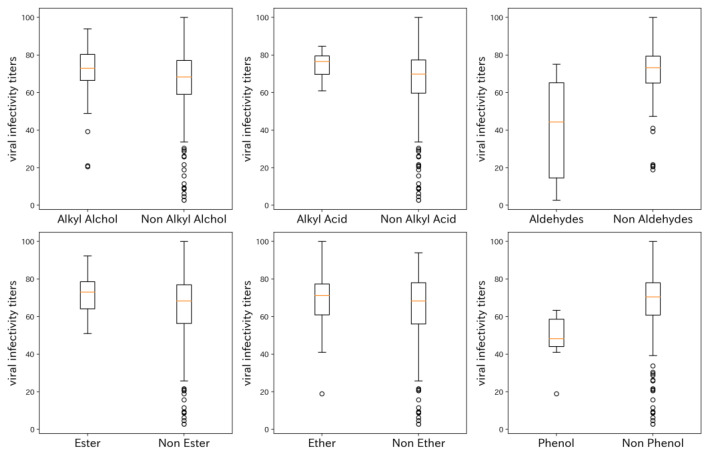
The box plot comparing viral infectivity titers by substructures. Orange lines indicate median values of the groups. A lower viral infectivity titer indicates a higher antiviral efficacy.

**Figure 2 microorganisms-12-01568-f002:**
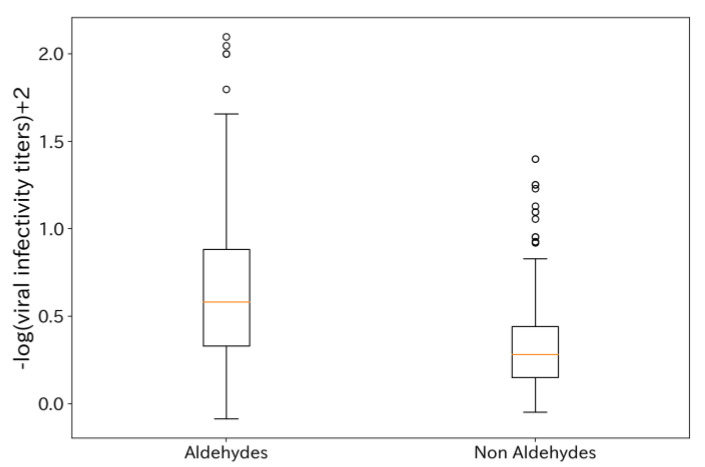
Box plot comparing viral infectivity titers of formulations composed of two types of fragrances with and without aldehyde compounds. Orange lines indicate median values of the groups. A higher −log(viral infectivity titer) + 2 indicates a higher antiviral efficacy.

**Figure 3 microorganisms-12-01568-f003:**
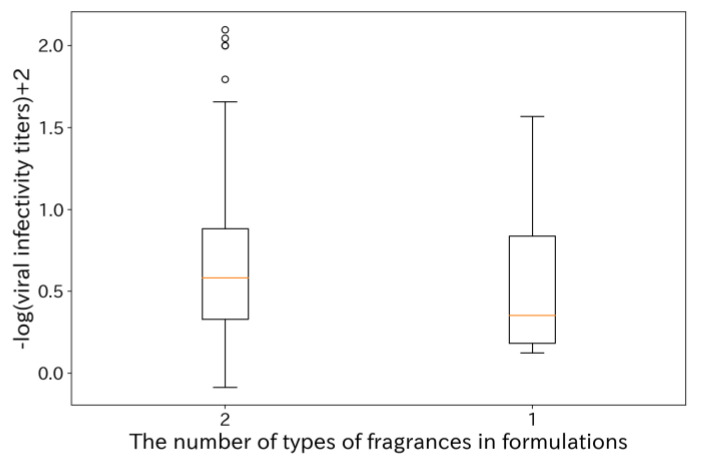
Box plot comparing viral infectivity titers between single aldehyde compounds and formulations containing aldehydes plus other compounds. Fragrance compound concentrations were uniformly evaluated at 0.1 vol%. For formulations containing two types of compounds, the concentrations of each compound were equally allocated. The orange line in the center of each box plot represents the median value for the group. A higher −log(viral infectivity titer) + 2 indicates a higher antiviral efficacy.

**Figure 4 microorganisms-12-01568-f004:**
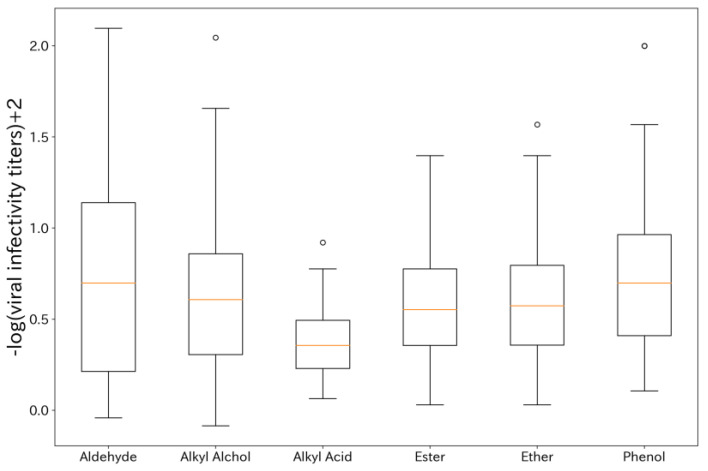
Box plots of viral infection titers for each compound combined with aldehyde compounds. The orange line in the center of each box plot represents the median value for the group. A higher −log(viral infectivity titer) + 2 indicates a higher antiviral efficacy.

**Figure 5 microorganisms-12-01568-f005:**
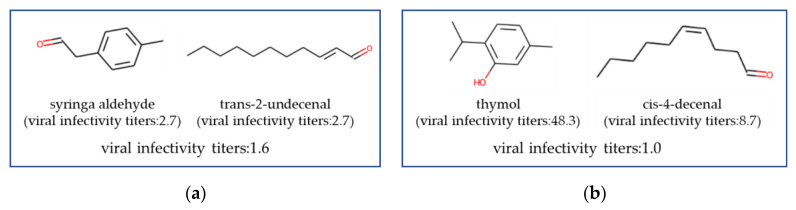
Highly effective antiviral formulations with two types of fragrance molecules. The viral infective titers represent the experimental value using fragrance formulation, whereas the viral infective titers in parentheses indicate the experimental values when using a single fragrance molecule alone. A lower viral infectivity titer indicates a higher antiviral efficacy. (**a**) viral infectivity titers: 1.6 (**b**) viral infectivity titers: 1.0.

**Figure 6 microorganisms-12-01568-f006:**
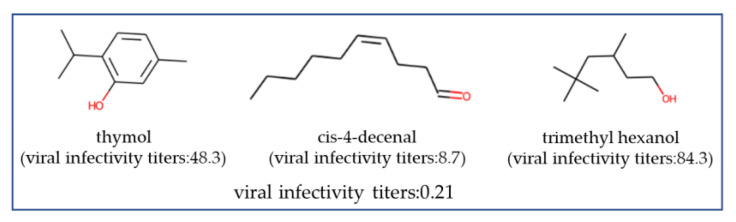
Highly effective antiviral formulations with three types of fragrance molecules. The viral infective titer represents the experimental value using the fragrance formulation, whereas the viral infective titers in parentheses indicate the experimental values obtained using single fragrance molecules alone. A lower viral infectivity titer indicates a higher antiviral efficacy.

**Figure 7 microorganisms-12-01568-f007:**
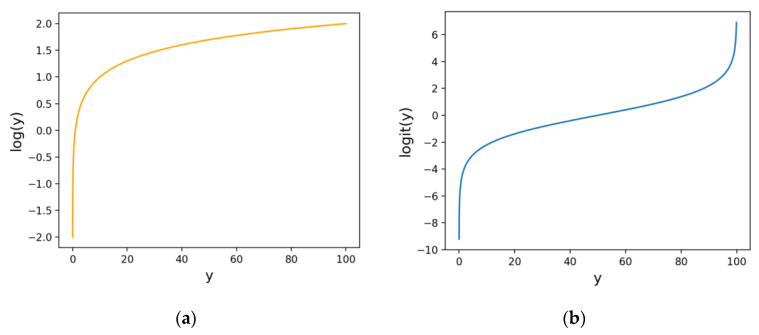
Logarithmic and logit functions. (**a**) Logarithmic function and (**b**) logit function.

**Figure 8 microorganisms-12-01568-f008:**
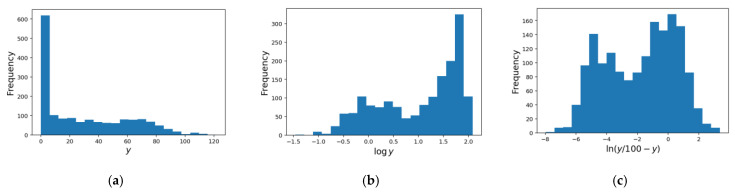
Viral infectious titer histograms. (**a**) Without preprocessing; (**b**) log transformation was performed; (**c**) logit transformation was performed.

**Figure 9 microorganisms-12-01568-f009:**
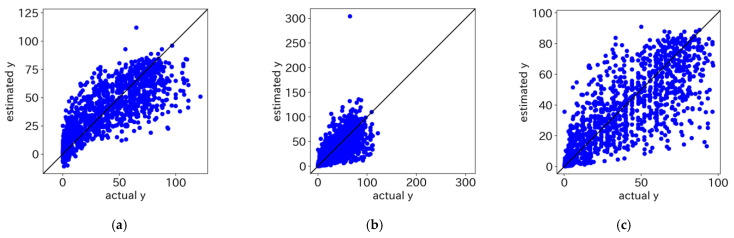
Y-Y plots for all predictions by GPR with $k_{1.5}$ kernel. (**a**) Without preprocessing; (**b**) log transformation was performed; (**c**) logit transformation was performed. The term “actual y” refers to the values obtained from experiments, while “estimated y” refers to the values computed using GPR models.

**Figure 10 microorganisms-12-01568-f010:**
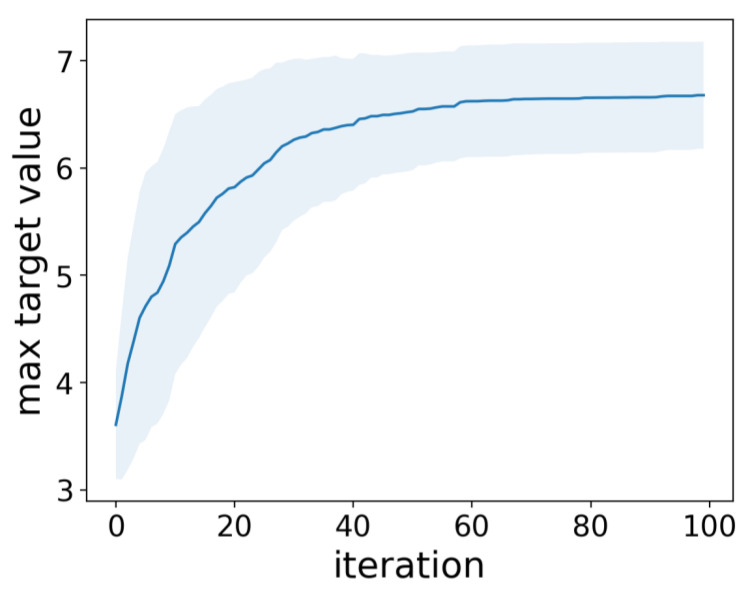
Performance of Bayesian optimization for viral infection titers using statistical descriptors. Mean max target value in each iteration (blue) and the region within ±1 standard deviation. (light blue).

**Figure 11 microorganisms-12-01568-f011:**
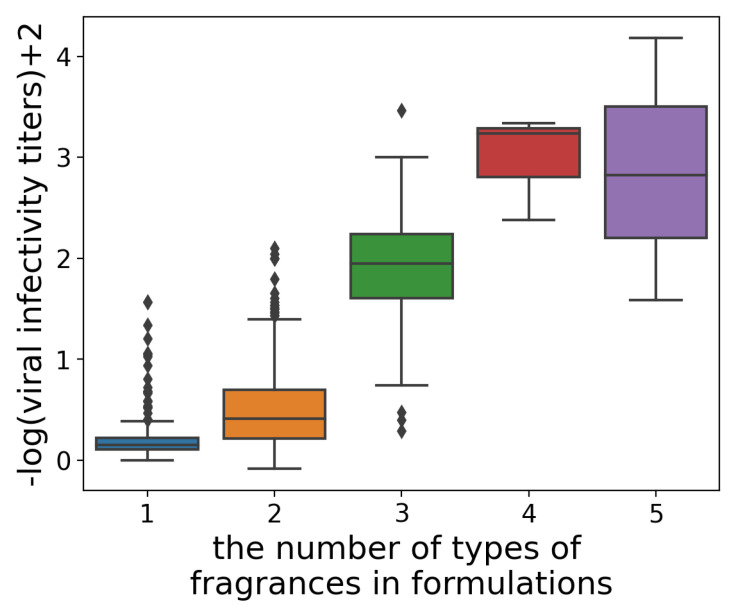
The box plot of viral infectivity titers for each type of fragrance formulation. A higher −log(viral infectivity titer) + 2 indicates a higher antiviral efficacy.

**Figure 12 microorganisms-12-01568-f012:**
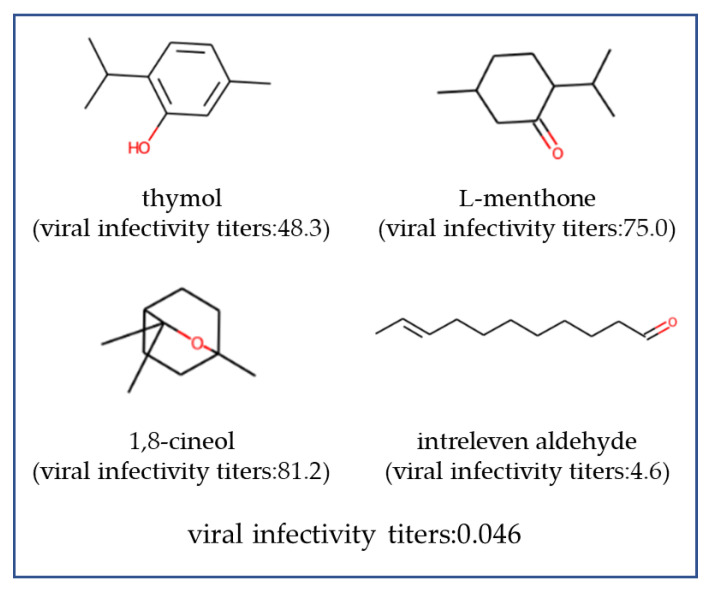
Highly effective antiviral formulations with four types of fragrance molecules. The viral infective titers represent the experimental value using fragrance formulation, whereas the viral infective titers in parentheses indicate the experimental values when using a single fragrance molecule alone. A lower viral infectivity titer indicates a higher antiviral efficacy.

**Figure 13 microorganisms-12-01568-f013:**
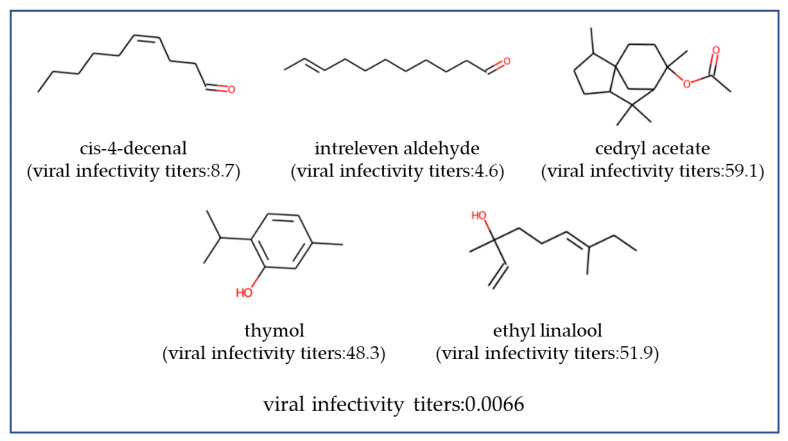
Highly effective antiviral formulations with five types of fragrance molecules. The viral infective titers represent the experimental value using fragrance formulation, whereas the viral infective titers in parentheses indicate the experimental values when using a single fragrance molecule alone. A lower viral infectivity titer indicates a higher antiviral efficacy.

**Table 1 microorganisms-12-01568-t001:** The performance of the GPRs using weighted means as statistical descriptors.

GPRs						
Kernel	$k_{\mathrm{RBF}}$	$k_{\mathrm{RBF}}$	$k_{\mathrm{RBF}}$	$k_{1.5}$	$k_{1.5}$	$k_{1.5}$
Transform	-	log	logit	-	log	logit
$R_{\mathrm{cv}}^{2}$	0.69	0.46	0.63	0.71	0.54	0.68
${\mathrm{RMSE}}_{\mathrm{cv}}$	16.6	22.1	17.6	16.1	20.4	16.2
${\mathrm{MAE}}_{\mathrm{cv}}$	11.7	13.4	11.0	11.3	12.2	10.2
${\mathrm{MAPE}}_{\mathrm{cv}}$	281	90.7	96.6	251	85.7	91.2

**Table 2 microorganisms-12-01568-t002:** The performance of the GPRs using weighted means and weighted standard deviations as statistical descriptors.

GPRs						
Kernel	$k_{\mathrm{RBF}}$	$k_{\mathrm{RBF}}$	$k_{\mathrm{RBF}}$	$k_{1.5}$	$k_{1.5}$	$k_{1.5}$
Transform	-	log	logit	-	log	logit
$R_{\mathrm{cv}}^{2}$	0.72	0.56	0.70	0.72	0.60	0.71
${\mathrm{RMSE}}_{\mathrm{cv}}$	15.9	19.9	15.9	15.8	19.0	15.6
${\mathrm{MAE}}_{\mathrm{cv}}$	11.1	12.1	9.93	11.0	11.6	9.73
${\mathrm{MAPE}}_{\mathrm{cv}}$	247	90.1	93.6	233	83.7	89.1

**Table 3 microorganisms-12-01568-t003:** The performance of the GPRs using weighted mean, weighted standard deviation, and maximum and minimum values as statistical descriptors.

GPRs						
Kernel	$k_{\mathrm{RBF}}$	$k_{\mathrm{RBF}}$	$k_{\mathrm{RBF}}$	$k_{1.5}$	$k_{1.5}$	$k_{1.5}$
Transform	-	log	logit	-	log	logit
$R_{\mathrm{cv}}^{2}$	0.72	0.54	0.70	0.72	0.58	0.72
${\mathrm{RMSE}}_{\mathrm{cv}}$	15.9	20.4	15.7	15.8	19.5	15.3
${\mathrm{MAE}}_{\mathrm{cv}}$	11.0	12.0	9.76	10.9	11.6	9.57
${\mathrm{MAPE}}_{\mathrm{cv}}$	232	93.6	92.6	221	85.5	88.6

## Data Availability

Data are contained within the article and Appendix A.

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
