# Peer review of "Design of Fragrance Formulations with Antiviral Activity Using Bayesian Optimization"

_microorganisms, 2024, doi:10.3390/microorganisms12081568_

Round 1

Reviewer 1 Report

Comments and Suggestions for Authors

The authors used Bayesian optimization, guided by Gaussian Process Regression (GPR), to identify formulations with antiviral efficacy. In general, the study is interesting from a methodological point of view, since the concentrations of aromatic substances used in the manuscript will still produce a noticeable odor. Moreover, these compounds are unlikely to be more effective than surfactants used in disinfectants.

Below are the reviewer's questions:

Abstract: «Confronted with the significant combinatorial challenge and the complexity of the compound formulation space, we have employed Bayesian optimization, guided by Gaussian Process Regression (GPR), to systematically explore and identify compounds with demonstrable antiviral efficacy» should be «Confronted with the significant combinatorial challenge and the complexity of the compound formulation space, we have employed Bayesian optimization, guided by Gaussian Process Regression (GPR), to systematically explore and identify FORMULATIONS with demonstrable antiviral efficacy»

1. Usually in medicinal chemistry papers, in particular in publications on antiviral compounds, the authors also evaluate the cytotoxicity of the studied compounds for host cells. Could the cytotoxicity of fragrances affect the results of Focus Forming Assay?

2. Please, provide the composition of the combinations of compounds described in paragraph 3.4

3. Table S1 contains repetitions, for example, geraniol, rose oxide

Line 17: of certain fragrance compound mixtures  - of mixtures of certain fragrance compounds

Line 116: «After the reaction, the virus was diluted with SFM and infected with MDCK cells (derived from canine renal tubular epithelial cells) previously cultured in 12 or 48 well plates.»

Most likely the cells were infected with the virus, and not vice versa

Line 242: delete AND in «with aldehydes or phenolic compounds AND were»

Line 247:  «a combination of syringa aldehyde, which has a formyl group at the benzyl position»

Please check the structure of suringa aldehyde. 3,5-dimethoxy-4-hydroxybenzaldehyde? You most likely named the compound with CAS 104-09-6 incorrectly

In many places, the subscript and superscript are not used (7.8 × 105 FFU, CO2)

Author Response

Thank you very much for taking the time to review this manuscript.

Comment1:

The authors used Bayesian optimization, guided by Gaussian Process Regression (GPR), to identify formulations with antiviral efficacy. In general, the study is interesting from a methodological point of view, since the concentrations of aromatic substances used in the manuscript will still produce a noticeable odor. Moreover, these compounds are unlikely to be more effective than surfactants used in disinfectants.

Response1:

Thank you for your feedback. While the formulations identified in this study may not have an exceptional fragrance, we can refine them in the future to improve both antiviral activity and scent. Although surfactants are indeed effective against viruses, ionic compounds such as sodium hypochlorite cannot be dispersed in the air and thus cannot be used to inactivate airborne viruses. Therefore, we believe that selective use of these compounds is feasible.

Comment2:

Below are the reviewer's questions:

Abstract: «Confronted with the significant combinatorial challenge and the complexity of the compound formulation space, we have employed Bayesian optimization, guided by Gaussian Process Regression (GPR), to systematically explore and identify compounds with demonstrable antiviral efficacy» should be «Confronted with the significant combinatorial challenge and the complexity of the compound formulation space, we have employed Bayesian optimization, guided by Gaussian Process Regression (GPR), to systematically explore and identify FORMULATIONS with demonstrable antiviral efficacy»

Response2:

Thank you for your valuable feedback. We have corrected the issue you mentioned. (Line 21)

Comment3:

  1. Usually in medicinal chemistry papers, in particular in publications on antiviral compounds, the authors also evaluate the cytotoxicity of the studied compounds for host cells. Could the cytotoxicity of fragrances affect the results of Focus Forming Assay?

Response3:

Thank you for your valuable feedback on our manuscript. 

We understand the importance of conducting the suggested experiment. However, we did not perform this experiment for the following reasons:

The formulation exhibiting approximately a 2-log reduction in antiviral activity was assessed by counting foci after dilution by more than 1000-fold. We believe the concentration was sufficiently low to avoid cytotoxic effects on MDCK cells. Additionally, in cases where cytotoxicity is observed, cell detachment can be seen in wells with 10-fold or 100-fold dilutions (concentration of each chemical in the formulation is 100 times or 10 times higher than 1000-fold dilution). However, in the current evaluation, such detachment was not observed, indicating that the reduction in the number of foci is not due to cytotoxic effects.

Following your suggestion, we added information on our manuscript that cell detachment was not observed during the focus forming assay in line 124-133.

Comment4:

  1. Please, provide the composition of the combinations of compounds described in paragraph 3.4

Response4:

Thank you for your feedback. We have included the molecular structures of the formulations exhibiting high antiviral activity, which were identified using Bayesian optimization, in the manuscript.(Figure 12 and 13)

Comment5:

  1. Table S1 contains repetitions, for example, geraniol, rose oxide

Response5:

Thank you for your valuable feedback. We have reviewed and corrected Table S1 as suggested. However, in this study, we conducted experiments using products with the same main components but different purities. Therefore, in our machine learning dataset, we organized the data based on the antiviral activity of the compound with the highest purity of the main component. As a result, there are compounds, such as Rose oxide, that appear multiple times in the dataset.

Comment6:

Line 17: of certain fragrance compound mixtures  - of mixtures of certain fragrance compounds

Response6:

Thank you for your valuable feedback. We have corrected the formatting issues you mentioned. (Line 17)

Comment7:

Line 116: «After the reaction, the virus was diluted with SFM and infected with MDCK cells (derived from canine renal tubular epithelial cells) previously cultured in 12 or 48 well plates.»

Most likely the cells were infected with the virus, and not vice versa

Response7:

Thank you for your insightful comment.

We have made the following revision based on your feedback in line 117-118.

“After the reaction, the virus was diluted with SFM and used to infect MDCK cells (derived from canine renal tubular epithelial cells) that were previously cultured in 12 or 48 well plates.

Comment8:

Line 242: delete AND in «with aldehydes or phenolic compounds AND were»

Response8:

Thank you for your feedback. Based on your suggestion, we have revised the manuscript by deleting the word "AND" from the phrase «with aldehydes or phenolic compounds AND were». We appreciate your careful review and valuable comments. (Line 252)

Comment9:

Line 247:  «a combination of syringa aldehyde, which has a formyl group at the benzyl position»

Please check the structure of suringa aldehyde. 3,5-dimethoxy-4-hydroxybenzaldehyde? You most likely named the compound with CAS 104-09-6 incorrectly

Response9:

Thank you for your feedback. We evaluated the compound using syringa aldehyde from Givaudan, as indicated in the following URL. Therefore, the structure is correct.
https://www.givaudan.com/fragrance-beauty/eindex/syringa-aldehyde-50

Comment10:

In many places, the subscript and superscript are not used (7.8 × 105 FFU, CO2)

Response10:

Thank you for your valuable feedback. We have carefully reviewed the manuscript and corrected the formatting issues you mentioned. (Line 113, 121)

Reviewer 2 Report

Comments and Suggestions for Authors

The manuscript on the design of fragrance formulations with antiviral activity using Bayesian optimization has several strengths, but there are also areas that could be improved to enhance its robustness and comprehensiveness before publishing:

 While the manuscript discusses the types of fragrances used and their combinations there are not enough information’s and there is a lack of specific detail regarding the concentration levels of each fragrance within the mixtures. There is not enough data about their source and purity.

The safety and potential toxicity of the fragrance mixtures at the concentrations used for antiviral activity are not sufficiently addressed. Since fragrances can sometimes cause allergic reactions or other health issues, this topic is lacking in the article

Lack of discussion on the mechanism of how these fragrances exert their antiviral effects.

The study does not address the long-term efficacy and stability of fragrance formulations. I could not identify how this are proposed to be used to reach the effective dose.

A comparative analysis with other existing antiviral agents (chemical or natural) is missing.

Comments on the Quality of English Language

English ok

Author Response

Thank you very much for taking the time to review this manuscript.

Comment1:

The manuscript on the design of fragrance formulations with antiviral activity using Bayesian optimization has several strengths, but there are also areas that could be improved to enhance its robustness and comprehensiveness before publishing:

 While the manuscript discusses the types of fragrances used and their combinations there are not enough information’s and there is a lack of specific detail regarding the concentration levels of each fragrance within the mixtures. There is not enough data about their source and purity.

Response1:

Thank you for your feedback. For formulations containing two or three types of compounds, the concentrations of each compound were equally allocated. For instance, in a formulation composed of compounds A and B, each was prepared at 0.05 vol%; similarly, for a formulation consisting of compounds A, B, and C, each was prepared at 0.033 vol%. Formulations with the composition A, A, B are also designated as formulations with three types of fragrances within the scope of this research.

Many of the fragrances were purchased from reagent manufacturers such as FUJIFILM Wako Pure Chemical Corporation, Tokyo Chemical Industry Co., Ltd., and Sigma-Aldrich. For those not available from these reagent manufacturers, we sourced them from companies like Bedoukian Research, Inc., International Flavors & Fragrances Inc., and Givaudan S.A.

Comment2:

The safety and potential toxicity of the fragrance mixtures at the concentrations used for antiviral activity are not sufficiently addressed. Since fragrances can sometimes cause allergic reactions or other health issues, this topic is lacking in the article.

Response2:

Thank you for your precise feedback regarding the lack of discussion on safety aspects, such as the potential for allergy induction, when using fragrances as antiviral agents in the Discussion section.

Using machine learning, our study identified a formulation that can induce antiviral activity at low concentrations. The formulation identified in this study has the potential for practical applications in real-life settings, such as diffusers, sprays, or sustained-release products.

However, we have not measured the concentration of the formulated compounds in the air, nor we have evaluated the concentrations at which antiviral activity can be achieved in the vapor phase. Therefore, as future research steps, it will be necessary to measure vapor-phase concentrations, design diffusion methods to achieve this (product development), control concentrations, and evaluate antiviral activity in the vapor phase.

By understanding the vapor-phase concentrations dependent on diffusion methods, it will be possible to assess allergy risks and evaluate the long-term efficacy and stability of fragrance formulations. We have added this as a necessary area of future research in the latter part of the Discussion section.(Line 502-513)

Thank you once again for your valuable insights, which have greatly contributed to enhancing the quality of our manuscript.

Comment3:

Lack of discussion on the mechanism of how these fragrances exert their antiviral effects.

Response3:

Thank you for your feedback. Investigating the mechanism by which compounds inactivate viruses is challenging, as it requires examining only the interaction between the virus and the compound. Therefore, when attempting to understand how a formulation with multiple compounds inactivates viruses, the level of difficulty significantly increases. We have documented the above content in the manuscript. (Line 514-528)

However, we have already started addressing this challenge and plan to discuss it in our next paper.

Comment4:

The study does not address the long-term efficacy and stability of fragrance formulations. I could not identify how this are proposed to be used to reach the effective dose.

A comparative analysis with other existing antiviral agents (chemical or natural) is missing.

Response4:

Thank you very much for your insightful comments. As for the long-term efficacy and stability of fragrance formulations, as mentioned in our previous response, these aspects depend on the design of diffusion methods (product development) and concentration control, and our study did not extend to their evaluation. Therefore, we have added explanations addressing your concerns, including the assessment of allergy risks, to the Discussion section in line 500-513.

Although not included in this paper due to the preliminary results, we have observed that achieving effective concentrations in the vapor phase requires both the volatility and water solubility of the compounds. Volatility is crucial for reaching sufficient spatial concentrations, and water solubility is necessary for dissolving in the moisture of virus-containing aerosols or droplets and acting on the virus.

While we are still gathering data and thus cannot discuss this in detail in this manuscript, we have confirmed that certain formulations show antiviral activity in the vapor phase at concentrations that pose no inhalation safety risks. We plan to accumulate more data on this in future studies.

Regarding the comparison with other existing antiviral agents, compounds with known antiviral activity, such as ethanol, do not exhibit effects at low concentrations. It has been reported that among 62 essential oils used at 10 mg/mL, three exhibited more than 53% anti-influenza virus activity (https://www.ncbi.nlm.nih.gov/pmc/articles/PMC6296812/pdf/ophrp-09-348.pdf). However, these evaluations were conducted at higher concentrations than our study's 0.1% concentration, resulting in less than a 1-log reduction in antiviral activity. Although not volatile, sodium hypochlorite is a representative compound with known antiviral activity, with effective concentrations ranging from 0.02% to 0.1% according to various reports. Additionally, antiviral activity has been evaluated for Triton X-100&TNBP solutions, with efficacy observed at concentrations of 0.3% to 1.0% (https://www.ajicjournal.org/article/S0196-6553(10)00187-2/abstract). Comparing the antiviral activity of these compounds with our findings, our formulation demonstrates effectiveness within a similar concentration range, suggesting potential use as a liquid formulation. However, considering the characteristics of fragrance components, we believe they are particularly suitable for applications in aerosol control in the vapor phase.

We have described a comparison with other existing antiviral agents in the Discussion section in line 407-436. Also we added a comparison of antiviral activity with essential oils in line 489-495.

Thank you once again for your valuable feedback.

Round 2

Reviewer 1 Report

Comments and Suggestions for Authors

The authors answered all my questions and comments

Author Response

Comment1:The authors answered all my questions and comments

Response1:

Thank you for your feedback. The statements regarding the potential for practical use in infection control in the Conclusion section were not supported by the data presented in the Results section. Therefore, we have revised these statements to indicate that they represent future possibilities in line 551-553.

Reviewer 2 Report

Comments and Suggestions for Authors

Although the authors did not cover all the points that we addressed, considering time limitations, I`m glad to see more information about the study that improves the overall understanding 

Author Response

Comments1:

Although the authors did not cover all the points that we addressed, considering time limitations, I`m glad to see more information about the study that improves the overall understanding 

Response1:

Thank you for providing us with the opportunity to address our shortcomings and improve the manuscript. We sincerely appreciate your constructive feedback and the chance to make necessary revisions.

Regarding the Introduction, we reviewed the previous revision comments but were unable to identify specific points that needed to be addressed. We believe that the comment may have been related to the lack of information on other existing antiviral agents, and have therefore added relevant details to address this issue in line 62-70. If our interpretation of the comments is incorrect, we would appreciate further clarification and are happy to make additional revisions as necessary.

We have revised the Methods section to include detailed information on the sources of the compounds and their purity in line 90-95, 99.